# SCD5 Regulation by VHL Affects Cell Proliferation and Lipid Homeostasis in ccRCC

**DOI:** 10.3390/cells12060835

**Published:** 2023-03-08

**Authors:** Athina Ganner, Antonia Philipp, Simon Lagies, Laura Wingendorf, Lu Wang, Felicitas Pilz, Thomas Welte, Kelli Grand, Soeren S. Lienkamp, Marinella Klein, Bernd Kammerer, Ian J. Frew, Gerd Walz, Elke Neumann-Haefelin

**Affiliations:** 1Renal Division, Department of Medicine, Medical Center—University of Freiburg, Faculty of Medicine, University of Freiburg, 79106 Freiburg, Germany; 2Core Competence Metabolomics, Hilde-Mangold-Haus, University of Freiburg, 79104 Freiburg, Germany; 3Institute of Organic Chemistry, University of Freiburg, 79104 Freiburg, Germany; 4Institute of Anatomy, Faculty of Medicine, University of Zurich, 8057 Zurich, Switzerland; 5BIOSS, Centre for Biological Signalling Studies, University of Freiburg, 79104 Freiburg, Germany; 6Spemann Graduate School of Biology and Medicine (SGBM), University of Freiburg, 79104 Freiburg, Germany; 7Department of Internal Medicine I, Medical Center—University of Freiburg, Faculty of Medicine, University of Freiburg, 79106 Freiburg, Germany

**Keywords:** ccRCC, VHL, SCD5, fat-7, HIF, ceramide

## Abstract

Clear cell renal cell carcinoma (ccRCC) is the most common histological subtype of renal cancer, and inactivation of the VHL tumor suppressor gene is found in almost all cases of hereditary and sporadic ccRCCs. CcRCC is associated with the reprogramming of fatty acid metabolism, and stearoyl-CoA desaturases (SCDs) are the main enzymes controlling fatty acid composition in cells. In this study, we report that mRNA and protein expression of the stearoyl-CoA desaturase SCD5 is downregulated in *VHL*-deficient cell lines. Similarly, in *C. elegans vhl-1* mutants, FAT-7/SCD5 activity is repressed, supporting an evolutionary conservation. SCD5 regulation by VHL depends on HIF, and loss of SCD5 promotes cell proliferation and a metabolic shift towards ceramide production. In summary, we identify a novel regulatory function of VHL in relation to SCD5 and fatty acid metabolism, and propose a new mechanism of how loss of VHL may contribute to ccRCC tumor formation and progression.

## 1. Introduction

Clear cell renal cell carcinoma (ccRCC) is the most common histological subtype of renal cell carcinoma (RCC), and is one of the ten most frequently occurring tumors in adults. The von Hippel-Lindau (*VHL*) tumor suppressor gene is mutated as an early event in almost all cases of ccRCC. The VHL protein targets the hypoxia-inducible transcription factor (HIF) for proteasomal degradation. This pathway is conserved in evolution between *Caenorhabditis elegans* (*C. elegans*) and mammals [1]. Inactivation of *VHL* in ccRCC leads to constitutive stabilization and activation of HIFα transcription factors, which coordinate numerous cellular activities that contribute to tumor formation and progression [2].

Activation of fatty acid (FA) biosynthesis is a universal metabolic feature of cancer cells, given their requirement as metabolic intermediates for membrane biosynthesis and energy storage. Moreover, FAs have important roles as signaling molecules, and alterations in the FA composition can affect a wide variety of cellular functions, including cell migration and induction of angiogenesis. Thus, FA balance is crucial for cancer cell growth and progression [3]. Stearoyl-CoA desaturases (SCDs) are key regulators of lipid homeostasis, by catalyzing the rate-limiting step in the biosynthesis of monounsaturated FAs (MUFA) from saturated FAs (SFA). In humans, two SCD isoforms, SCD1 (noted as SCD) and SCD5, have been identified. SCD is expressed ubiquitously, while SCD5 localizes to specific tissues.

SCD is upregulated in the majority of cancers, including ccRCC, and participates in cell metabolism, cell cycle progression, and tumor cell migration [4,5]. SCD upregulation was also linked to tumor aggressiveness and poorer prognosis in ccRCC [6]. Inhibition of SCD has been shown to limit essential lipid metabolites and proliferation, and induces apoptosis in ccRCC cell models [7,8]. While SCD has been extensively investigated, the role of SCD5 in metabolism and disease is still poorly defined. The expression of SCD5 is restricted to few tissues, including kidneys, supporting a tissue-specific function [9,10]. A growing body of evidence indicates that decreased SCD5 activity is specific to some types of cancer, and correlates with their aggressiveness and poor patient prognosis. In advanced melanoma, where SCD5 is downregulated, its re-expression modified extracellular matrix proteins and reversed the epithelial-mesenchymal-like process [11,12]. These observations suggest an anti-neoplastic activity of SCD5, particularly during metastatic dissemination. A recent analysis of breast cancer samples showed that low SCD5 expression was associated with more aggressive cancer phenotypes and shorter patient survival [9].

In this study, we investigated the biological function of SCD5 in ccRCC. We found that SCD5 expression was strongly downregulated in both primary ccRCC tissue samples, as well as in ccRCC cell lines. VHL deficiency and cellular hypoxia caused the suppression of SCD5 expression in an HIF-dependent manner. We found that in *C. elegans*, loss of *vhl-1* resulted in downregulation of the SCD-5 ortholog *fat-7*, underscoring the evolutionary conservation of our observation. Furthermore, we observed that inhibition of SCD5 causes lipidomic profile changes, and favors cell proliferation.

## 2. Materials and Methods

### 2.1. Reagents and Plasmids

PT2385 was obtained from Abcam, Cambridge, UK, and used at concentrations as indicated. SCD5 was subcloned by PCR from human cDNA (Agilent, Santa Clara, CA, USA), and fused by standard cloning techniques to a pcDNA6 vector encoding an N-terminal Flag-tag (Invitrogen, Carlsbad, CA, USA), and to a pLXSN vector with a Venus tag, a variant of the yellow fluorescent protein (YFP).

### 2.2. Antibodies

Antibodies used in this study included antibody to SCD5 (Aviva systems biology, San Diego, CA, USA, SCD5 antibody-C-terminal region, ARP65045_P050, 1:1000 dilution), antibody to VHL (Cell Signaling, Leiden, The Netherlands, 1:1000 dilution), antibody to β-actin (Sigma, Darmstadt, Germany), antibody to γ-tubulin, and antibody to HIF2α from Abcam, Cambridge, UK.

### 2.3. Cell Culture and Transfections

Cell culture and transfections were carried out as previously described [13]. In brief, HEK 293T and HeLa cells were grown in Dulbecco’s modified Eagle’s medium (DMEM) supplemented with 10% FBS. RPTEC cells were obtained from ATCC (Manassas, VA, USA) and cultured in DMEMF12 supplemented with hTERT Immortalized RPTEC Growth Kit (ATCC) and G418. VHL-negative 786-O and RCC4 cells, as well as RCC4 cells expressing VHL, were provided by Ian Frew [14]. The 786-O cells were cultured in RPMI1640 medium supplemented with 10% FBS, and RCC4 cells were cultured in DMEM supplemented with 10% FBS and Geniticin (0.5 mg/mL). For 293T cells, transient transfections were carried out using the calcium phosphate method; cells were lysed 24 h after transfection. For expression of SCD5 in RCC4 cells, cells were transduced with Venus.SCD5 in pLXSN. To analyze SCD5 protein levels after hypoxia treatment, RPTEC cells were grown in a hypoxia chamber (0.5% oxygen, overnight), while control cells were grown under standard conditions.

For quantification of the levels of endogenous proteins of interest, cells were split in parallel and lysed in a buffer containing 20 mM Tris-HCl (pH 7.5), 1% Triton X-100, 50 mM NaF, 15 mM Na_4_P_2_O_7_, 0.1 mM EDTA, 50 mM NaCl, 2 mM Na_3_VO_4_, and cOmplete protease inhibitor (Roche, Basel, Switzerland). For Figure 4c, cells were lysed in a buffer containing 50mM Tris (pH 8.0), 120 mM NaCl, 0.5% NP-40, 0.01mM EDTA, 10% glycerol, and cOmplete protease inhibitor (Roche). The total amount of solubilized proteins was determined by the Bradford method (Bio-Rad, Hercules, CA, USA), and equal amounts of total proteins were subjected to SDS-PAGE and immunoblotting analysis. Blots were scanned and bands were quantified using LabImage 1D 4.1 software. Data were expressed as means or as means ± SEM. Statistical analysis was performed using GraphPad Prism 9 software.

### 2.4. shRNA Mediated Knockdown (Stable Polyclonal Cell Lines)

To generate a HeLa cell line for tetracycline-inducible knockdown of VHL, as well as a RCC4 cell line for tetracycline-inducible knockdown of SCD5, a lentivirus-based transduction system (pLVTH) was used as previously described [15]. For knockdown induction, cells were treated with tetracycline for 72 h (5 µg/mL). The 786-O SCD5 knockdown cell line constitutively expresses SCD5 shRNA. shRNA targeting sequences were: control targeting sequence (ctrl) 5′ gtacgcggaatacttcga 3′, VHL shRNA targeting sequence 5′ ctg gtt aac caa act gaa tta tt 3′, and SCD5 shRNA targeting sequence 5′ ctc tat tct ccg cta ta ccat ct 3′.

### 2.5. Quantitative Real Time PCR

Total RNA was obtained from indicated cell lines using RNeasy Mini Kit (Qiagen, Hilden, Germany) and reverse-transcribed using the SuperScript™ IV First-Strand Synthesis System (Invitrogen) according to the manufacturer’s protocol. qRT–PCR was performed on a LightCycler 480 (LC 480, Roche). GAPDH, HSPCB, or β-actin were used as normalization controls as indicated. Each biological replicate was measured in technical triplicates. Significant differences were assessed by Student’s unpaired *t*-test. The primers used for qRT–PCR were: VHL: 5′CACAGCTACCGAGGTCAC3′ and 3′CTGAATTATTTGTGCCATCTCTCA5′, HSPCB: 5′TCTGGGTATCGGAAAGCAAGCC3′ and 3′CAAGATGCCTGAGGAAGTGCAC5′, GAPDH: 5′CATTTCCTGGTATGACAA3′ and 3′CAAGAGGAAGAGAGAGAC5′, PAI-1: 5′ CCTGGTTCTGCCCAAGTTCT3′ and 3′ATCGAGGTGAACGAGAGTGG5′, VEGF: 5′TACCTCCACCATGCCAAGTG3′ and 3′GAAGGAGGAGGGCAGAATCAT5′, β-actin: 5′TCCTTCCTGGGCATGGAG3′ and 3′TGAAGTGTGACGTGGACATC5′, SCD5 primer_1: 5′GGCAACCAAGCCGATGATC3′ and 3′ACTTGGAACAGCCATCCCAC5′, SCD5 primer_2: 5′GAGGAATGTCGTCCTGATGAGC3′ and 3′CCTACTTCTGCTTCCTCCTGGC5′, SCD: 5′TCGGATATCGTCCTTATGACAAGA3′ and 3′CTGTGGGTGAGGGCTTCC5′. The quantification of changes in mRNA expression levels was based on the 2^−ΔΔCt^ method.

### 2.6. Gene Expression Analysis in C. elegans

All strains were maintained at 20 °C without starvation prior to experiment. Total RNA purified from age-synchronized young adult *C. elegans* was prepared using TRI Reagent (Sigma-Aldrich, St. Louis, MO, USA) and an RNA clean and concentrator kit (Zymo Research Corp., Irvine, CA, USA). DNase treatment was performed using the on-column DNase digestion (Qiagen, Hilden, Germany). Reverse transcription was performed by SuperScript IV (Invitrogen) and oligo-dT primer. Quantitative PCR was performed on a Roche LightCycler 480 using the Takyon No Rox SYBR MasterMix blue dTTP (Eurogentec, Seraing, Belgium). qPCR reactions were performed in at least three independent samples in triplicates. Samples were analyzed by the 2^−ΔΔCt^ method, with normalization to the geometric mean of the reference genes cdc-42 and Y45F10D.4. Significant differences were assessed by Student’s unpaired *t*-test. Primer sequences were: fat-5: 5′TCCAGAGGAAGAACTACC3′ and 3′TCATTCCAGTCGTCCTCT5′, fat-7: 5′GCGCTGCTCACTATTTTGGT3′ and 3′CGTTGGTGAAGGAGGTCACA5′, cdc-42: 5′CTGCTGGACAGGAAGATTACG3′ and 3′CTTCATTCGAGAATGTCCGAG5′, Y45F10D.4:5′ GTCGCTTCAAATCAGTTCAGC3′ and 3′GTCGGATCACTTGACAAGAAC5′.

### 2.7. RNA Sequencing and Data Analysis

Total RNA from three biological replicates each, of *vhl-1* mutants and wild type animals, was extracted from age-synchronized young adult *C. elegans.* Library preparation and RNA sequencing were conducted by GATC Biotech AG, Ebersberg, Germany, on an Illumina platform with single-end 50 bp mRNA sequencing. Sequencing data were analyzed on the Galaxy platform [16]. Quality check was assessed with FastQC v0.71 (http://www.bioinformatics.babraham.ac.uk/projects/fastqc/ accessed on 10 October 2018). Trim Galore! version 0.4.3.1 was used for trimming (http://www.bioinformatics.babraham.ac.uk/projects/trim_galore/ accessed on 10 October 2018), STAR v2.6.0b for alignment, featureCounts v1.6.3 for counting the reads, and DESeq2 v2.2.12.40.3 for differential expression analysis. For significantly differentially expressed genes, a cut-off of p-adj < 0.05 was used. For visualization, the Volcano plot for differences between *vhl-1* and wild-type animals was drawn using R software 4.1.3.

### 2.8. Data Availability

RNA-seq datasets have been deposited in GEO (Gene Expression Omnibus) under the accession number # GSE224016.

### 2.9. Cell Viability Assay

At 6 h after transient transfection, 293T cells were counted and seeded in at least triplicates in 96-well plates (approx. 5 × 10^4^ cells/well). RCC4 and 786-O cells were counted and seeded 24 h after last splitting (approx. 1 × 10^4^ cells/well). RCC4 cells expressing Venus.SCD5, and 293T cells expressing F9.SCD5, were cultured in medium without FBS, before analysis of viability. Viability was assessed using the MTT Cell Proliferation Assay Kit (Cayman Chemical, Ann Arbor, MI; USA), performed according to the manufacturer’s protocol. Viability was calculated relative to control.

### 2.10. C. elegans Growth Conditions

Unless otherwise indicated, *C. elegans* were cultured at 20 °C using standard methods. The strains used were as follows: Wild-type N2. CB5602 *vhl-1(ok161).* ZG31 *hif-1(ia4).* CB6090 *hif-1(ia4);vhl-1(ok161).* DMS303 *nIs590[Pfat-7::fat-7::GFP + lin-15(+)].* ENH687 *nIs590[Pfat-7::fat-7::GFP];vhl-1(ok161).* BX113 *lin-15B* and *lin-15A(n765)*;*waEx15[Pfat-7::GFP + lin15(+)]*. ENH699 *vhl-1(ok161);waEx15[Pfat-7::GFP + lin15(+)].* ENH695 *hif-1(ia4);vhl-1(ok161);waEx15[Pfat-7::GFP + lin15(+)].* ENH697 *hif-1(ia4);waEx15[Pfat-7::GFP + lin15(+)].*

### 2.11. Microscopy and Quantification of FAT-7::GFP Expression

Light microscopy was performed using a Zeiss Axio Imager M2 microscope equipped with Nomarski differential interference contrast (DIC), an AxioCam camera, and the ZEN Blue 2.3 pro software. Worms were raised on NGM until L4/young adult stage at 20 °C, unless otherwise listed. For microscopy, worms were placed on 2% agarose pads on slides and stunned with 8mM levamisole. GFP was detected using an EGFP-filter set (480/20 nm excitation, 510/20 nm emission). All image data intended for quantitative comparison were acquired at the same sub-saturating exposure time. FAT-7::GFP fluorescence was quantified in FIJI/ImageJ. Green fluorescence intensities were background subtracted and normalized to mean value in control worms.

### 2.12. Hypoxia Treatment

To examine FAT-7::GFP expression after exposure to hypoxia, L3 animals were placed in a hypoxia chamber (Billups-Rothenberg, Inc., San Diego, CA, USA) at 0.5% O_2_ balanced by N_2_ for 16 h at 20 °C. Control animals were exposed to room air (21% O_2_) for 16 h. Imaging of FAT-7::GFP was performed as described previously above.

### 2.13. Metabolomics Analysis

Untargeted GC/MS and targeted LC/MS lipid profiling was conducted as reported previously [17]. In brief, cells were washed with 0.9% NaCl and quenched by 1 mL methanol:water (1:1). Cells were lysed by vortexing after the addition of 0.5 mL chloroform. Dried extracts were derivatized by methoxyamination and silylation for gas-chromatography coupled to electron ionization mass-spectrometry. Metabolites were identified by matching retention index and spectral fingerprints to a commercially available and an in-house database. Lipid extracts were reconstituted and analyzed by liquid-chromatography coupled to multiple reaction monitoring mass spectrometry. Pooled quality control samples were prepared by mixing equal amounts of each sample and regularly injected within the sequence. Analytes were normalized to internal standards and the sum of all annotated metabolites/lipids, to account for differences in biological mass.

### 2.14. Statistical Analyses

Data were expressed as means or as means ± SEM and analyzed using GraphPad Prism 9 Software. Differences between the experimental groups were evaluated using unpaired Student’s *t*-test (comparisons between subjects) or one-way ANOVA (comparisons across more than two groups). Metabolomics data were analyzed by MetaboAnalyst 5.0 [18], including principal component analysis and unpaired *t*-test, followed by false discovery rate correction and heat map generation. Two technical replicates from two independent experiments were used.

## 3. Results

### 3.1. SCD5 Expression in ccRCC

To determine the clinical significance of SCD5 for ccRCC, we queried the Cancer Genome Atlas (TCGA), CPTAC databases [19] (http://ualcan.path.uab.edu, accessed on 9 June 2021), and the human protein atlas [20] (Human Protein Atlas proteinatlas.org, accessed on 9 June 2021) for SCD5 expression levels in patient samples. SCD5 mRNA expression was significantly downregulated in primary ccRCC samples compared to control kidney samples (Figure 1a). Analysis of the CPTAC mass-spectrometry-based proteomic tumor dataset showed that SCD5 protein levels were significantly lower in primary kidney cancer samples compared to control kidney samples (Figure 1b). Consistent with these observations, immunohistochemistry revealed lower SCD5 expression in tumor tissues compared to control kidney tissues (Figure 1c). Next, we analyzed SCD5 expression in *VHL*-deficient ccRCC cell lines. SCD5 expression was significantly lower in the *VHL*-deficient RCC4 and 786-O ccRCC cell lines compared to normal human renal proximal tubular epithelial cells (RPTECs; Figure 1d). To gain further insight into the clinical significance of SCD5 for ccRCC, we explored the correlation between SCD5 mRNA expression and patient outcome. Downregulation of SCD5 expression was associated with shorter overall survival (Figure 1e). Collectively, these data indicate that SCD5 expression is differentially regulated in ccRCC, and downregulation of SCD5 expression may have a prognostic clinical value.

### 3.2. SCD5 Expression Depends on VHL

Next, we investigated the regulatory mechanisms underlying *SCD5* downregulation associated with ccRCC development. Inactivation of the *VHL* tumor suppressor gene is the signature event in ccRCC. To evaluate the impact of VHL on SCD5, HeLa cells were infected with lenti-*VHL* shRNA or lenti-control shRNA. shRNA-mediated *VHL* knockdown efficacy was verified by RT-qPCR and Western blot analysis (Figure 2a,b). In *VHL*-depleted cells, *SCD5* mRNA levels were significantly downregulated (Figure 2c). To further investigate the impact of VHL, we examined *SCD5* mRNA expression in ccRCC cells. Reconstitution of VHL resulted in significant upregulation of *SCD5* expression compared to *VHL*-deficient RCC4 control cells (Figure 2d). Moreover, re-introduction of VHL in RCC4 cells strongly increased SCD5 protein levels compared to RCC4 cells (Figure 2e). Thus, reduced SCD5 expression in ccRCC cells depends on the loss of VHL function.

### 3.3. VHL Regulates SCD5 Expression in C. elegans

To investigate the cellular pathways controlled by *VHL*, we took advantage of the *C. elegans* model system. We and others have previously shown that *C. elegans* is ideally suited to study the complexity of the VHL pathway [13,21]. Components of the VHL-HIF pathway are evolutionarily conserved in *C. elegans*: as in mammals, the mutation of *vhl-1* leads to stabilization of the transcription factor HIF-1, and induction of transcriptional programs promoting bioenergetic adaption to hypoxia [1].

RNA sequencing analysis in *C. elegans vhl-1* null mutants compared to wild type animals revealed a downregulation of *fat-5* and *fat-7* (Figure 3a), two close orthologs of human SCD5. Comparison of the amino acid sequence showed that *C. elegans fat-7* and *fat-5* share 42.8% and 40.9% sequence identity with human SCD5, respectively (pairwise sequence alignment by EMBOSS Needle). RT-qPCR in *vhl-1* mutants confirmed reduced expression of *fat-5* and *fat-7* mRNA compared to wild-type (Figure 3b). Due to the greater sequence homology between SCD5 and *fat-7*, we focused on *fat-7* in the following experiments.

We studied the expression of *fat-7* in vivo using a transgenic reporter expressing a translational fusion of *fat-7* to GFP. *vhl-1* mutants showed a significant reduction of FAT-7::GFP expression, while FAT-7::GFP was broadly expressed in wild-type worms (Figure 3c). Inactivation of *vhl-1* leads to induction of a hypoxia transcription program. To address whether *fat-7* expression is also affected by hypoxic conditions, wild-type animals were treated in a hypoxic chamber at 0.5% oxygen. Exposure to low oxygen levels resulted in a strong reduction of FAT-7::GFP expression compared to normoxia conditions (Figure 3d).

The key role of VHL in the regulation of HIF transcription factors is well characterized [22]. In addition, numerous HIF-independent functions of VHL have been described [22]. To assess whether the expression of *fat-7* depends on HIF-1, the *fat-7*::GFP transgenic reporter was introduced into *hif-1* mutants and *hif-1;vhl-1* double mutants. While inactivation of *vhl-1* consistently repressed *fat-7* expression, a strong loss of function mutation in *hif-1* resulted in a significant upregulation of FAT-7::GFP expression (Figure 3e). Interestingly, *hif-1;vhl-1* double mutants showed a slightly higher FAT-7::GFP expression compared to *hif-1* mutants (*p* = 0.032; Figure 3e). Quantification of *fat-7* expression by RT-qPCR confirmed these results (Figure 3f). The induction of *fat-7* transcript level was slightly more pronounced in *hif-1;vhl-1* double mutants, suggesting that knockdown of hif-1 abrogated not only negative effects of *vhl-1* mutation on fat-7 expression, but might also free up positive regulatory mechanisms. In addition to HIF-1, alternative factors may act in parallel and influence the expression of fat-7 (Figure 3f).

Together, these experiments support an important role of VHL-1/HIF-1 in *fat-7* transcriptional regulation.

### 3.4. SCD5 Expression in ccRCC Cells Depends on HIF2

Our results indicate that HIF-1 regulates *fat-7* expression in *C. elegans*. We sought to further investigate this regulatory mechanism in ccRCC cells. Therefore, ccRCC cells were treated with the HIF2α inhibitor PT2385. PT2385 binds to HIF2α and prevents it from binding to DNA, thus inhibiting its transcriptional activity, as evidenced by decreased expression of the target genes *PAI-1* and *VEGF* (Figure 4a). We observed that treatment with the HIF2α inhibitor significantly upregulated the mRNA level of *SCD5* in 786-O cells (Figure 4a). Next, we measured the protein levels of SCD5 following treatment with PT2385. In *VHL*-defective 786-O and RCC4 cells, inhibition of HIF2α resulted in upregulation of SCD5 protein levels (Figure 4b,c). Together these findings confirm the HIF2 dependence of SCD5 expression in ccRCC cells. Additionally, SCD5 protein levels were reduced in RPTEC cells grown under hypoxic conditions (Figure 4d).

### 3.5. SCD5 Affects Cell Physiology and Metabolism

As SCD5 was downregulated in ccRCC, we hypothesized that its loss could affect cell proliferation. To test this, we introduced shRNA-mediated knockdown of *SCD5* in RCC4 and 786-O cells. *SCD5* knockdown efficiency was verified by RT-qPCR and Western blot analysis (Figure 5a–d). *SCD5* knockdown resulted in significantly increased proliferation rates (Figure 5e,f). Conversely, overexpression of SCD5 in HEK 293T and RCC4 cells significantly reduced cell proliferation (Figure 5g). Together these observations suggest an inhibitory role of SCD5 on cell proliferation.

Cancer cells differ from normal cells by distinct biological hallmarks, including the reprogramming of metabolic processes. We undertook an unbiased approach to assess metabolic and lipidomic alterations by SCD5 in ccRCC cells. Using gas chromatography-mass spectrometry (GC/MS), a metabolic fingerprint was generated in 786-O cells following *SCD5* knockdown, and compared to 786-O cells.

The data set was first analyzed by principal component analysis (Appendix A). Statistical analysis revealed 12 significantly altered metabolites. These significantly altered endometabolites are displayed in a heat map showing range-scaled z-scores (Figure 6a). Distinct clustering is visible between *SCD5* knockdown cells and control. In *SCD5* knockdown cells, downregulated metabolites included key amino acids, such as glutamic acid and alanine, and some sugars (glucose 6-phosphate, fructose, threitol).

Next, the lipophilic extracts were analyzed by targeted liquid chromatography-mass spectrometry (LC/MS). Marked differences were observed in the principal component analysis of 786-O cells upon *SCD5* knockdown compared to control (Appendix A). A total of eight lipids were significantly altered in *SCD5* knockdown cells (Figure 6b). Four ceramides were upregulated in *SCD5*-deficient cells compared to control. As a bioactive lipid, ceramide has been implicated in a variety of physiological functions, including apoptosis, cell growth arrest, cell migration, and adhesion. Specifically, ceramide has been shown to accumulate in ccRCC patients irrespective of the tumor grade, whereas globosides decreased in a grade-dependent manner in ccRCC [23]. In *SCD5*-deficient cells Lactosylceramides (Lac-cer) and Hexosylceramid (Hex-ceramid), which belong, among others, to the globosides, were found to be downregulated (Figure 6b). Taken together, loss of SCD5 in ccRCC cells alters lipid metabolism towards increased ceramide production, and downregulation of globosides.

## 4. Discussion

Our study investigated the role of SCD5 in ccRCC, starting from its downregulation in human specimen and in ccRCC cell lines. Survival analysis revealed that low SCD5 expression was related to prognosis in ccRCC. VHL deficiency and cellular hypoxia caused suppression of SCD5 expression in an HIF-dependent manner. Importantly, our *C. elegans* analyses demonstrate that the link between VHL/HIF and SCD5 is conserved from nematodes to human. Furthermore, SCD5 seems to be involved in oncogenic transformation and cell proliferation.

### 4.1. Regulation of SCD5 Expression

The endogenous levels of SCD5 mRNA expression and protein levels were reduced in ccRCC tumor samples (Figure 1). The in vivo significance of this finding was corroborated by reduced SCD5 expression in *VHL*-defective ccRCC cells, and *VHL* shRNA knockdown cells (Figure 1d and Figure 2c). Conversely, transient over-expression of VHL resulted in a concomitant increase of SCD5 expression in ccRCC cells (Figure 2d,e). It is well established that inactivation of VHL results in stabilization of HIF2, and activation of its target genes, such as VEGF and PAI, for angiogenesis and proliferation [22]. SCD5 is also subjected to regulation by HIF2 in *VHL*-deficient ccRCC cells, as pharmacological inhibition of HIF2α in RCC4 and 786-O cells restored SCD5 expression (Figure 4). Together, these data show that the canonical VHL-HIF2 pathway regulates SCD5 expression in ccRCC.

Remarkably, the mechanism of SCD5 regulation by the VHL-HIF pathway is conserved in evolution between nematodes and mammals. Consistent with the suppression of SCD5 in *VHL*-deficient cells, *C. elegans vhl-1* null mutants displayed reduced *fat-7* (ortholog of SCD5) mRNA expression by qPCR (Figure 3). Similarly, *fat-7* expression was reduced under hypoxic conditions (Figure 3). Of note, HIF becomes stabilized under hypoxia, as well as loss of VHL function. In line with canonical VHL-HIF signaling, *fat-7* was significantly upregulated in *hif-1* mutants (Figure 3). Taken together, human and *C. elegans* data indicate a negative regulatory effect of HIF on SCD5/*fat-7* expression.

*SCD5* is regulated primarily on the transcriptional level by several different hormones (insulin, leptin), nutrients, and transcription factors (including SREBP1, EGR2, and PPAR) [24]. How could HIF2 limit SCD5 activity? HIF transcriptional activity can be repressed via post-translational modifications, and protein-protein interactions that block the binding of HIFα subunits to ARNT/HIFβ and p300/CBP [25]. Hydroxylation of HIF by FIH-1, or phosphorylation by casein kinase 1, inhibit HIF transcriptional activity [26]. Alternatively, HIF2 could regulate a transcriptional repressor. HIF has been shown to induce the transcriptional repressor DEC1, which inhibits adipogenesis by repressing PPARy [27]. In addition, microRNAs (miRs) have been shown to regulate SCD5 expression in melanoma cells. SCD5 mRNA is targeted by the oncogenic miR-221 and miR-222, and this correlates with melanoma progression [12]. The mechanism of HIF-mediated inhibition of SCD5 expression remains to be determined in future studies.

### 4.2. Role of SCD5 in Metabolism and Cell Proliferation

A large body of evidence has determined the role of SCD1 as an oncogenic factor, and also that the synthesis of MUFA is essential for cancer cells [28]. A growing number of studies suggest the possibility that SCD5 also has a role in the modulation of mitogenic and tumorigenic events. Analysis of breast cancer samples showed that low expression of SCD5 was associated with more aggressive cancer phenotypes, and correlates with the prognosis of breast cancer. The most affected biological functions by SCD5 were involved in negative regulation of the cell cycle and cell division [9]. It was shown, in a metastatic cell model for breast cancer, that over-expression of SCD5 decreased extracellular matrix deposition and hampered metastatic spreading [29].

Studies in melanoma found that SCD5 could protect against metastatic progression. Advanced melanomas down-regulate SCD5. Enforced expression of SCD5 in melanoma cells inhibits the secretion of extracellular matrix proteins, with the consequent impairing of tumor spreading [11]. Re-expression of SCD5 drove advanced melanoma cells toward differentiation, and reversed the epithelial-mesenchymal process [12]. In our study, we explored the role of SCD5 in ccRCC cell proliferation. While knockdown of SCD5 accelerated the rate of 786-O and RCC4 cell proliferation, SCD5 over-expression induced a reduction in cell proliferation (Figure 5e–g).

Our lipidomic data suggest that SCD5 deficiency leads to disruption of lipid homeostasis, resulting in accumulation of ceramides and downregulation of globosides (Figure 6b). Alteration of the fatty acid balance has a profound impact on the biophysical properties of all lipids, including membrane integrity, and has also been shown to influence ER stress and cell death [30]. Interestingly, accumulation of ceramide in ccRCC cell lines has been shown to induce NF-κB signaling, which might contribute to cancer development [31]. Together, our data suggest that low SCD5 activity contributes to ccRCC malignancy by modulation of lipogenic pathways, and is implicated in oncogenic proliferation. Of note, previous bioinformatic analyses identified SCD5 as a biomarker for ccRCC. SCD5 was significantly downregulated in ccRCC, and was incorporated in a prognostic risk score model that can predict the overall survival time of ccRCC patients [32,33].

Our observations, in conjunction with previous studies, suggest that the two SCD isoforms might play separate roles in the mechanisms of cancer. Studies that more comprehensively address the role of SCD5 in cell- and tissue-specific cancers are needed, to gain insight into the involvement of SCD5 in metabolism and tumorigenesis.

## 5. Conclusions

In summary, our study shows that mRNA and protein levels of SCD5/*fat-7* are substantially downregulated in *VHL*-deficient ccRCC tumor samples and ccRCC cells, as well as in C. elegans *vhl-1* loss-of-function mutants. Exposure to hypoxic conditions resembles VHL loss in both systems, suggesting that regulation of SCD5/*fat-7* occurs via the conserved VHL-HIF pathway. Genetic *hif-1* inhibition in *C. elegans*, and pharmacological HIF2 inhibition in ccRCC cells, restored SCD5/*fat-7* activity. Decreased SCD5 expression promotes ccRCC proliferation and affects lipid homeostasis. Taken together, our data extend previous findings, suggesting that SCD5 plays a key role in tumor formation.

## Figures and Tables

**Figure 1 cells-12-00835-f001:**
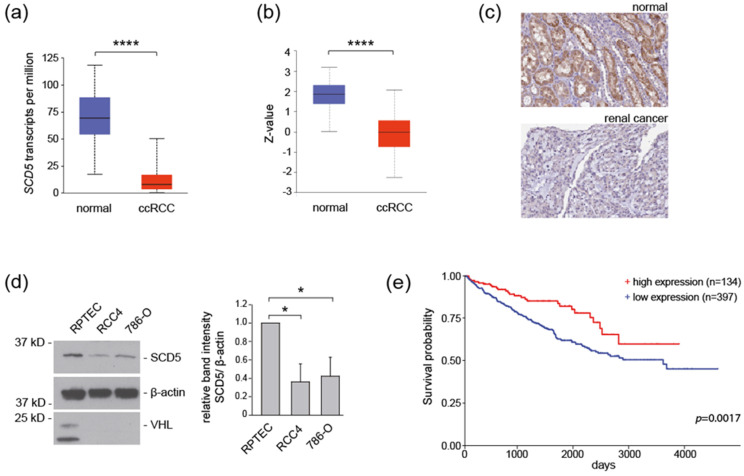
**SCD5 is downregulated in *VHL*-deficient ccRCC tumors and cell lines.** (**a**) Expression of *SCD5* mRNA in TCGA ccRCC (*n* = 533) and normal kidney samples (*n* = 72). **** *p* < 0.0001. The analysis was performed using the UALCAN/TCGA platform. (**b**) SCD5 protein expression is reduced in ccRCC (*n* = 110) compared to normal kidney samples (*n* = 84) (CPTAC dataset). **** *p* < 0.0001. (**c**) Representative result of SCD5 expression using renal cancer (https://www.proteinatlas.org/ENSG00000145284-SCD5/pathology/renal+cancer#img, accessed on 9 June 2021) and normal kidney samples (https://www.proteinatlas.org/ENSG00000145284-SCD5/tissue/kidney#img, accessed on 9 June 2021) from the Human Protein Atlas. (**d**) SCD5 protein levels are reduced in RCC4 and 786-O ccRCC cells compared to RPTEC. Cell lysates of RCC4, 786-O, and RPTEC cells were analyzed by Western blot using anti-SCD5 antibody. Protein levels of SCD5 were quantified using LabImage 1D software and normalized to β-actin protein levels. The right panel shows quantification of relative SCD5 protein levels from three independent experiments. Data are presented as mean ± SEM. * *p* < 0.05 (*t*-test). kD, kilodalton. Full-length blots are presented in Appendix A. (**e**) low SCD5 expression (*n* = 134) correlates with shorter overall survival (high SCD5 expression, *n* = 397). The analysis was performed using the UALCAN platform.

**Figure 2 cells-12-00835-f002:**
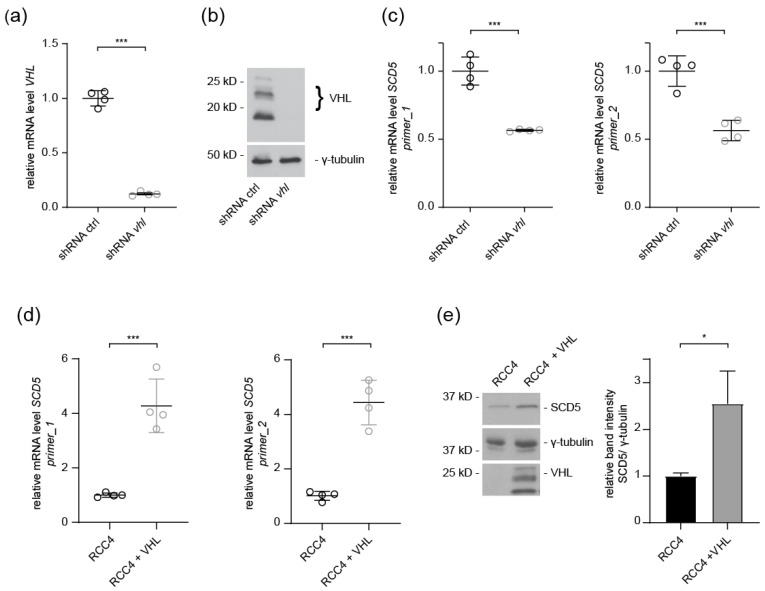
**SCD5 expression in ccRCC cells is VHL dependent.** (**a**) mRNA levels of *VHL* in HeLa cells infected with inducible control-shRNA or *VHL*-specific shRNA assayed by RT-qPCR; HSPCB was used as the reference gene. *n* = four. Data are presented as mean ± SEM. *** *p* < 0.001 (*t*-test). (**b**) Representative blot showing VHL expression in HeLa cells infected with lenti-control shRNA or lenti-VHL shRNA. Full-length blots are presented in Appendix A. (**c**) SCD5 mRNA levels are affected by VHL. qPCR analysis of HeLa cells expressing shRNA against control or VHL. Two different primer sets were used for *SCD5* mRNA analysis; HSPCB was used as the reference gene. *n* = four for each experiment. Data are mean ± SEM. *** *p* < 0.001 (*t*-test). (**d**) Analysis of *SCD5* mRNA levels in RCC4 cells and RCC4 cells expressing VHL. Cell lysates were prepared from *VHL*-deficient RCC4 cells and cell lines transduced with VHL (RCC4 + VHL); GAPDH was used as the reference. *n* = four for each experiment. Data are mean ± SEM. *** *p* < 0.001 (*t*-test). (**e**) Western blot demonstrating SCD5 expression in RCC4 cells (*VHL* deficient) and RCC4 cells transduced with VHL. Full-length blots are presented in Appendix A. Right panel shows quantification of relative SCD5 levels from three independent experiments. * *p* < 0.05 (*t*-test).

**Figure 3 cells-12-00835-f003:**
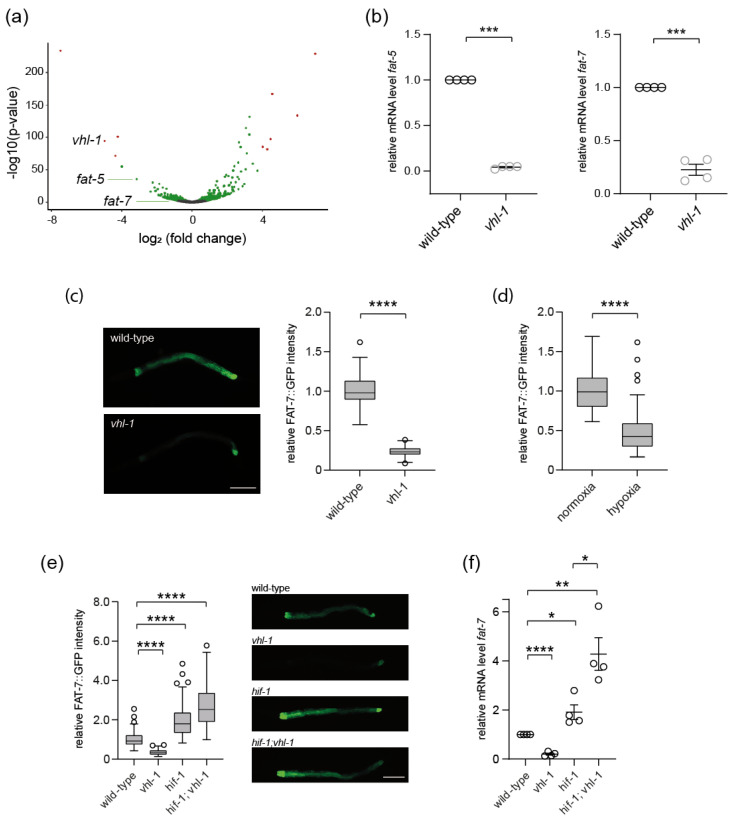
***fat-7*/SCD5 is downregulated in *C. elegans vhl-1* mutants.** (**a**) Volcano plot showing differentially expressed genes between *vhl-1* mutants and wild-type. (**b**) *fat-5* and *fat-7* expression is downregulated in *vhl-1* mutants. *fat-5 and fat-7* mRNA expression in wild-type animals and *vhl-1(ok161) mutants* was analyzed by RT-qPCR. Data are mean ± SEM of four independent experiments. *t*-test. *** *p* < 0.001. (**c**) *vhl-1* mutants show reduced FAT-7::GFP expression. Representative fluorescence images of otherwise wild-type and *vhl-1(ok161)* animals expressing a *fat-7*::GFP translational reporter (nIs590). Scale bar indicates 100 μm. Right panel shows quantification of FAT-7::GFP in wild-type and *vhl-1* mutants. Three independent experiments were performed. n > 50 for each condition. Mean ± SEM. **** *p* < 0.0001 (*t*-test). (**d**) Effect of hypoxia on FAT-7::GFP expression. Wild-type animals expressing FAT-7::GFP were exposed to low oxygen (0.5% in a hypoxia chamber) and compared to controls grown under standard conditions (21% oxygen). Quantification of FAT-7::GFP expression. n > 50 for each condition. Mean ± SEM. **** *p* < 0.0001 (*t*-test). (**e**) VHL-1 and HIF-1 antagonistically regulate *fat-7* expression. Fluorescence micrographs of wild-type, *hif-1(ia04)*, *vhl-1(ok161)*, and *hif-1;vhl-1* mutants showing expression of FAT-7::GFP. Scale bar indicates 100 μm. Left panel shows quantification of FAT-7::GFP fluorescence intensity. Three independent experiments, n > 50 for each genotype. Mean ± SEM. **** *p* < 0.0001 (ANOVA). (**f**) Inactivation of *hif-1* abrogates downregulation of *fat-7* expression in *vhl-1* mutants. RT-qPCR results showing *fat-7* mRNA levels in wild-type, *vhl-1(ok161)*, *hif-1(i104)*, and *hif-1;vhl-1* mutants. * *p* < 0.05, ** *p* < 0.01, **** *p* < 0.0001 (*n* = four for each genotype).

**Figure 4 cells-12-00835-f004:**
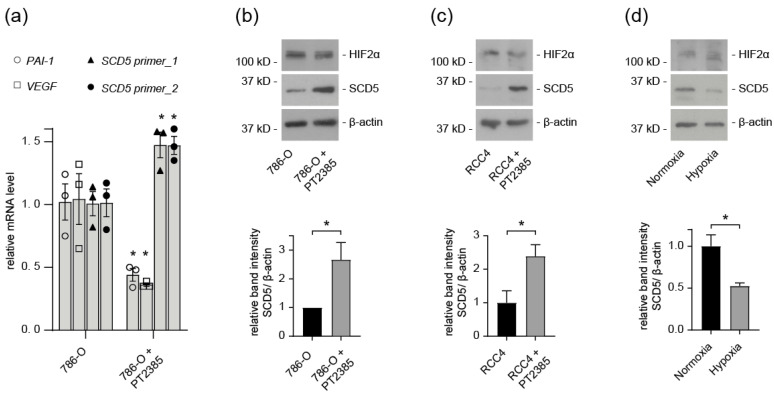
**SCD5 expression depends on HIF2α.** (**a**) 786-O cells were incubated with the HIF2α inhibitor PT2385 (1 µmolar) for 72 h. RNA expression of *SCD5*, *VEGF*, and *PAI-1* were assessed by RT-qPCR relative to HSPCB. *n* = three for each experiment. Data are represented as mean ± SEM. * *p* < 0.05 compared to untreated cells (*t*-test). (**b**) SCD5 protein levels are restored after treatment with the HIF2α inhibitor PT2385. The 786-O cells were incubated with the HIF2α inhibitor PT2385 (1 µmolar) or DMSO (control) for 72 h, and cell lysates were assessed by immunoblotting with anti-SCD5 and anti-HIF2α antibody. Quantification of three independent experiments. Data represent mean ± SEM. * *p* < 0.05 (*t*-test). kD, kilodalton. Full-length blots are presented in Appendix A. (**c**) Immunoblot of RCC4 cells that were treated with PT2385 (1 µM) or DMSO control. Quantification of relative SCD5 levels from three independent experiments. Data are presented as mean ± SEM. * *p* < 0.05 (*t*-test). Full-length blots are presented in Appendix A. (**d**) SCD5 protein levels after hypoxia treatment of RPTEC cells. Quantification from three independent experiments. Data represent mean ± SEM. * *p* < 0.05 (*t*-test). kD, kilodalton. Full-length blots are presented in Appendix A.

**Figure 5 cells-12-00835-f005:**
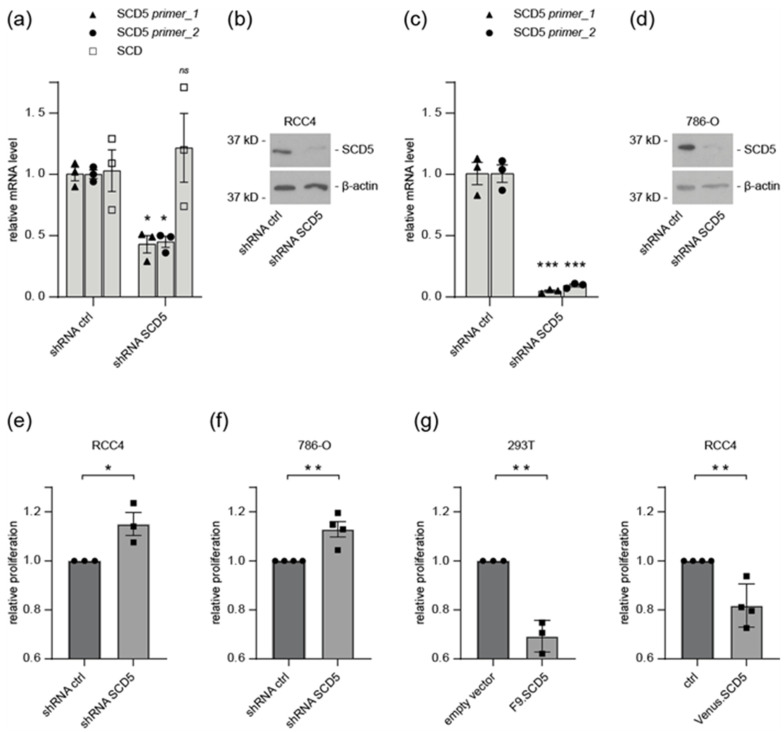
**SCD5 affects cell proliferation.** (**a**) RT-qPCR analysis of SCD5 and SCD in RCC4 cells infected with inducible control or *SCD5*-specific shRNA. Two different primer sets were used for *SCD5* mRNA analysis. SCD5 shRNA does specifically affect SCD5, as SCD was not down regulated. HSPCB was used as the reference gene, *n* = three. Data are presented as mean ± SEM. * < 0.05 (*t*-test). (**b**) Western blot of shRNA-mediated knockdown of *SCD5* in RCC4 cells. (**c**) mRNA levels of *SCD5* in 786-O cells infected with control-shRNA or *SCD5*-specific shRNA, assayed by qPCR; β-actin was used as the reference gene. *n* = three. Data are presented as mean ± SEM. *** *p* < 0.001 (*t*-test). (**d**) Western blot of shRNA-mediated knockdown of *SCD5* in 786-O. (**e**) Knockdown of SCD5 induces cell proliferation. Cell viability of RCC4 cells upon shRNA *SCD5* knockdown was assayed compared to control. Data are presented as mean ± SEM of three independent experiments. * *p* < 0.05 (*t*-test). (**f**) Cell viability of 786-O cells upon shRNA *SCD5* knockdown was assayed compared to control. Data are presented as mean ± SEM of three independent experiments. ** *p* < 0.01 (*t*-test). (**g**) Overexpression of SCD5 in 293T and RCC4 cells reduces cell proliferation. Cell viability was assayed in 293T and RCC4 cells transfected with Flag-tagged SCD5, or transduced with Venus-tagged SCD5, as indicated. Data are presented as mean ± SEM of three independent experiments. ** *p* < 0.01 (*t*-test).

**Figure 6 cells-12-00835-f006:**
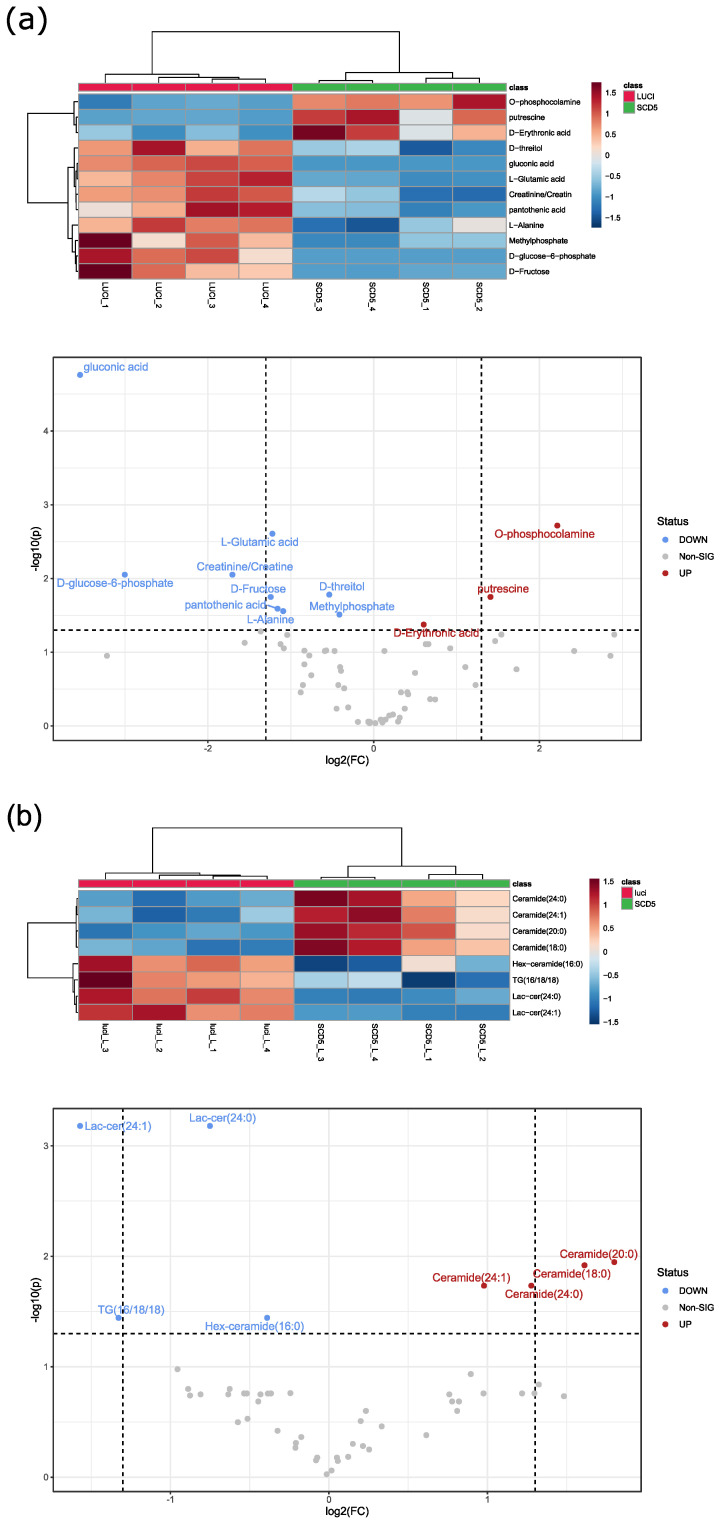
**Metabolic and lipidomic assessment of ccRCC cells following SCD5 knockdown.** (**a**) Heat map and volcano plot analysis of endometabolites in 786-O cells infected with control or *SCD5*-specific shRNA. Only significantly altered metabolites (unpaired *t*-test, corrected for multiple testing) are displayed. Cluster analysis after Euclidian and Ward are displayed on top for samples, and on the left side for metabolites. Red: control 786-O cells. Green: *SCD5* shRNA 786-O cells. Range scaled z-scores are displayed. *n* = four. (**b**) Heat map and volcano plot analysis of significantly altered lipids in 786-O cells infected with control or *SCD5*-specific shRNA. Significance was determined by *t*-test. Cluster analysis after Euclidian and Ward revealed marked differences in *SCD5* knockdown conditions. Red: control 786-O cells. Green: *SCD5* shRNA 786-O cells. Range scaled z-scores are displayed. *n* = four.

## Data Availability

RNA-seq datasets have been deposited in GEO (Gene Expression Omnibus) under the accession number # GSE224016.

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
