# Peer review of "SCD5 Regulation by VHL Affects Cell Proliferation and Lipid Homeostasis in ccRCC"

_cells, 2023, doi:10.3390/cells12060835_

Round 1
Reviewer 1 Report
The manuscript by Ganner et al. investigates the function of SCD5 in clear cell renal cell carcinoma (ccRCC) and the impact of VHL on SCD5 and its role in ccRCC tumor formation. The authors report that SCD5 gene expression is downregulated in VHL-deficient cell lines and is dependent on the HIF gene. The support the argument, the author also used the C. elegans model and ortholog of SCD5, fat-7 is downregulated in VHL-deficient C. elegans.
Major comments.
1. The authors suggest that VHL regulation of SCD5 depends on HIF, however, the Figure 3e-f shows that fat-7 (SCD5) is upregulated in the double knockout model suggesting alternative factors involved in fat-7 regulation or there is an indirect effect of VHL on SCD5. Also in Fig 3a. fat-7 is not marked in the volcano plot and the authors should discuss why they did not pursue the mutational analysis of fat-5 in C. elegans.
2. The authors should perform a pull-down assay to show that VHL physically interacts with SCD5.
Author Response
Response to Reviewer 1:
The manuscript by Ganner et al. investigates the function of SCD5 in clear cell renal cell carcinoma (ccRCC) and the impact of VHL on SCD5 and its role in ccRCC tumor formation. The authors report that SCD5 gene expression is downregulated in VHL-deficient cell lines and is dependent on the HIF gene. The support the argument, the author also used the C. elegans model and ortholog of SCD5, fat-7 is downregulated in VHL-deficient C. elegans.
Major comments.
- The authors suggest that VHL regulation of SCD5 depends on HIF, however, the Figure 3e-f shows that fat-7(SCD5) is upregulated in the double knockout model suggesting alternative factors involved in fat-7 regulation or there is an indirect effect of VHL on SCD5.
We thank the reviewer for this important comment. vhl-1;hif-1 double mutants showed an slightly increased fat-7 expression compared to hif-1 animals (assayed by qPCR and by in vivo transcriptional reporter fused to GFP). This observation is very interesting and suggests that knockdown of hif-1 abrogated not only negative effects of vhl-1 mutation on fat-7 expression, but might also free up positive regulatory mechanisms. We agree with the reviewer that in addition to HIF-1 alternative factors may act in parallel to HIF and influence the expression of fat-7. Several transcriptional regulators for fat-7 have been described in C. elegans including the nuclear hormone receptors NHR-49/HNF4A and NHR-80, SBP-1/SREBP, DAF-16/FOXO, and MDT-15/MED15. In addition, environmental conditions, such as temperature and food availability, influence fat-7. Our data support the involvement of VHL-HIF signaling in fat-7 regulation at the transcriptional level. The observation that the expression of fat-7 is largely altered in response to environmental conditions, together with the requirement of multiple transcriptional regulators, indicates that the precise regulation of fat-7 is crucial for lipid metabolism. These observations were discussed in more detail in the revised manuscript (page 8, lines 327-330).
Also in Fig 3a. fat-7 is not marked in the volcano plot and the authors should discuss why they did not pursue the mutational analysis of fat-5 in C. elegans.
We thank the reviewer for this important comment and apologize that we did not label fat-7 in the volcano plot. Fat-7 has now been marked in the figure.
By Alliance of Genome resources (https://www.alliancegenome.org) three C. elegans orthologs of human SCD5 have been proposed (fat-5, fat-6, fat-7) and characterized in detail by Brock et al., PLOS Genetics 2006. We observed that fat-6 expression was not regulated by VHL-1 or HIF-1 (data not shown), while vhl-1 mutants showed a strong downregulation of both, fat-5 and fat-7 to a comparable level. Comparison of the C. elegans and human Δ9 desaturases showed that fat-7 shares a higher sequence homology with human SCD5 (42,8% identity). Therefore we focused on fat-7 in our study (page 7, line 307-p.8, line 310).
- The authors should perform a pull-down assay to show that VHL physically interacts with SCD5.
Thank you for the suggestion. We have performed co-immunoprecipitation experiments with SCD5 and VHL as suggested by the reviewer. While we were able to show an interaction between SCD1 and VHL (data not shown as not the focus of the manuscript), there was no or at most a very discrete binding between SCD5 and VHL. We suggest that the regulation of SCD5 by VHL occurs via additional signaling factors at the transcriptional level, so we would not necessarily expect the two proteins to interact at the protein level.
Reviewer 2 Report
In the manuscript entitled “Analysis of SCD5 and the VHL pathway in ccRCC”, authors investigated the biological function of SCD5 in ccRCC.
The study is elegantly designed, in a way that very clearly leads to the elucidation of the given goal. The chosen methods are very diverse, modern and powerful and complement each other in the best way. What's more, the authors used different models, evolutionarily very distant, in order to confirm the existence of common mechanisms of regulation of the physiological properties of cells. In my opinion, methodologically and conceptually, this research represents a good whole, which provides readers with an insight into the importance of SCD5 on oncogenic transformation and cell proliferation, from different angles, from molecular mechanisms to physiological properties of cells. With all that, the authors effectively use the available literature to support the assumptions based on the obtained results.
My only suggestion would be that the authors try to summarize their results and assumptions in a clear conclusion that would emphasize the overall importance of this research. I think that the manuscript should end with such a conclusion. The last paragraph of the Discussion chapter, as it stands now, with its vagueness casts a shadow over a good study concept.
Author Response
Response to Reviewer 2:
In the manuscript entitled “Analysis of SCD5 and the VHL pathway in ccRCC”, authors investigated the biological function of SCD5 in ccRCC.
The study is elegantly designed, in a way that very clearly leads to the elucidation of the given goal. The chosen methods are very diverse, modern and powerful and complement each other in the best way. What's more, the authors used different models, evolutionarily very distant, in order to confirm the existence of common mechanisms of regulation of the physiological properties of cells. In my opinion, methodologically and conceptually, this research represents a good whole, which provides readers with an insight into the importance of SCD5 on oncogenic transformation and cell proliferation, from different angles, from molecular mechanisms to physiological properties of cells. With all that, the authors effectively use the available literature to support the assumptions based on the obtained results.
My only suggestion would be that the authors try to summarize their results and assumptions in a clear conclusion that would emphasize the overall importance of this research. I think that the manuscript should end with such a conclusion. The last paragraph of the Discussion chapter, as it stands now, with its vagueness casts a shadow over a good study concept.
We thank the reviewer for the suggestion. A chapter on conclusions has been introduced (page 15, lines 524-533).
- Conclusions
In summary, our study shows that mRNA and protein levels of SCD5/fat-7 are substantially downregulated in VHL-deficient ccRCC tumour samples and ccRCC cells, as well as in C. elegans vhl-1 loss-of-function mutants. Exposure to hypoxic conditions resembles VHL loss in both systems, suggesting that regulation of SCD5/fat-7 occurs via the conserved VHL-HIF pathway. Genetic hif-1 inhibition in C. elegans and pharmacological HIF2 inhibition in ccRCC cells restored SCD5/fat-7 activity. Decreased SCD5 expression promotes ccRCC proliferation and affects lipid homeostasis. Taken together, our data extend previous findings suggesting that SCD5 plays a key role in tumor formation.
Reviewer 3 Report
Dear editor and authors,
Thanks for having me review this manuscript “Analysis of SCD5 and the VHL pathway in ccRCC” submitted to the journal “Cells”. In this study, the authors reported SCD5 is downregulated in VHL-deficient cell lines. VHL regulated the expression of SCD5 depending on HIF. Loss of VHL causes the loss of SCD5 which contributes to ccRCC tumor progression. The regulatory mechanism between SCD5 and VHL was studied in human cell lines and C. elegans. To make this story clearer, some points should be further addressed.
Major points:
1. The title “Analysis of SCD5 and the VHL pathway in ccRCC” is too broad. Commonly, we use one concluded sentence as the title. The readers will be curious about the relationship between SCD5 and VHL and how they contribute to ccRCC.
2. As described in this study, VHL targeting HIF and HIF is involved in hypoxia. However, the relative expression level of fat-7 for hif-1 mutant and hypoxia was opposite in C. elegans model. The hypoxia condition was just tested in C. elegans model. How hypoxia affects SCD5 should be tested in the human cell line.
3. The authors studied HIF-1 in C. elegans model and HIF2 in the human cell line. Are they conserved in these two species?
Minor points:
1. Page 2 line 97, Na4P2O7, Na3VO4 should be revised.
2. Page 4 line 199, N2 should be revised.
3. The statistical significance in the figures is not uniform, some are shown as stars, and some are shown as p-value.
4. In the C. elegans model, the study focused on fat-7, which was not shown in figure 3a. In addition, the 2 in the X axis should be subscript.
5. Page 8 line 310, Reference should be added after “… have been described”.
Author Response
Response to Reviewer 3:
Thanks for having me review this manuscript “Analysis of SCD5 and the VHL pathway in ccRCC” submitted to the journal “Cells”. In this study, the authors reported SCD5 is downregulated in VHL-deficient cell lines. VHL regulated the expression of SCD5 depending on HIF. Loss of VHL causes the loss of SCD5 which contributes to ccRCC tumor progression. The regulatory mechanism between SCD5 and VHL was studied in human cell lines and C. elegans. To make this story clearer, some points should be further addressed.
Major points:
- The title “Analysis of SCD5 and the VHL pathway in ccRCC” is too broad. Commonly, we use one concluded sentence as the title. The readers will be curious about the relationship between SCD5 and VHL and how they contribute to ccRCC.
The title has been changed to: SCD5 regulation by VHL affects cell proliferation and lipid homeostasis in ccRCC
- As described in this study, VHL targeting HIF and HIF is involved in hypoxia. However, the relative expression level of fat-7 for hif-1 mutant and hypoxia was opposite in C. elegansmodel.
It is well established that loss of VHL-1 function or hypoxic conditions lead to stabilization of HIF transcription factors. This regulatory mechanism is conserved between C. elegans and mammals. Indeed, we observed in vhl-1 mutants and under hypoxic conditions that, through a HIF-dependent mechanism, fat-7 expression was repressed. By contrast, in hif-1(ia04) mutants we found that fat-7 expression was upregulated. hif-1(ia04) mutants bear a 1,231-bp deletion of the second, third, and fourth exon. This introduces a frameshift and premature stop in the mutant mRNA, and likely leads to a complete loss of hif-1 function (Jiang et al. PNAS 2001). Consistent with a repressive function of HIF-1 towards fat-7, the expression of fat-7 was relieved in hif-1 mutants.
The hypoxia condition was just tested in C. elegans model. How hypoxia affects SCD5 should be tested in the human cell line.
As suggested by the reviewer, we have added the analysis of SCD5 protein levels in RPTEC cells exposed to hypoxic conditions (Fig. 4d). Consistent with the results in the C. elegans model system, SCD5 protein levels were downregulated upon hypoxia.
- The authors studied HIF-1 in C. elegansmodel and HIF2 in the human cell line. Are they conserved in these two species?
Yes, HIF-1 in C. elegans and HIF2 in humans are conserved: In humans there are three HIFα proteins that are targeted by VHL, but HIF1α and HIF2α have been the most extensively studied (Kaelin and Ratcliffe, Molecular Cell 2008) and deregulation of particularly HIF2 drives ccRCC tumorigenesis (William G. Kaelin Jr., The Journal of Clinical Investigation 2022). In contrast to mammalian cells, C. elegans has a single HIFa homolog (HIF-1) (Jiang et al. PNAS 2001, Epstein et al., Cell 2001). C. elegans hif-1 may diverged from an ancestral HIFa gene before duplication events gave rise to mammalian isoforms. The conservation between HIF-1 and HIF-2a has been shown in Jiang et al., PNAS 2001.
Minor points:
- Page 2 line 97, Na4P2O7, Na3VO4 should be revised.
We apologize for this, the writing has been revised.
- Page 4 line 199, N2 should be revised.
Again, we apologize for this, the writing has been revised.
- The statistical significance in the figures is not uniform, some are shown as stars, and some are shown as p-value.
We are sorry for this; the statistical significances are now shown consistently as stars.
- In the C. elegansmodel, the study focused on fat-7, which was not shown in figure 3a. In addition, the 2 in the X axis should be subscript.
Thank you for the comment. fat-7 has been added to the Volcano Blot. The writing of log2 has also been revised.
- Page 8 line 310, Reference should be added after “… have been described”.
As suggested by the reviewer, a reference has been added.
Round 2
Reviewer 1 Report
All questions were answered.
Reviewer 3 Report
All the questions were addressed.